# Expression of Cry1Ab/2Aj Protein in Genetically Engineered Maize Plants and Its Transfer in the Arthropod Food Web

**DOI:** 10.3390/plants12234057

**Published:** 2023-12-02

**Authors:** Yi Chen, Michael Meissle, Jiabao Xue, Nan Zhang, Shulin Ma, Anping Guo, Biao Liu, Yufa Peng, Xinyuan Song, Yan Yang, Yunhe Li

**Affiliations:** 1Institute of Tropical Bioscience and Biotechnology, Chinese Academy of Tropical Agricultural Sciences, Haikou 571025, China; 2Hainan Yazhou Bay Seed Laboratory, Sanya 572025, China; 3Research Division Agroecology and Environment, Agroscope, 8046 Zurich, Switzerland; 4Key Laboratory of Genetics and Germplasm Innovation of Tropical Special Forest Trees and Ornamental Plants, Ministry of Education, College of Forestry, Hainan University & School of Tropical Agriculture and Forestry (School of Agricultural and Rural Affairs, School of Rural Revitalization), Hainan University, Haikou 570228, China; 5Nanjing Institute of Environmental Sciences, Ministry of Ecology and Environment, Nanjing 210042, China; 6Agro-Biotechnology Research Institute, Jilin Academy of Agricultural Sciences, Changchun 130033, China; 7Department of Environmental Systems Science, ETH Zurich, 8092 Zurich, Switzerland; 8State Key Laboratory for Plant Diseases and Insect Pests, Institute of Plant Protection, Chinese Academy of Agricultural Sciences, Beijing 100193, China; 9State Key Laboratory of Cotton Bio-Breeding and Integrated Utilization, School of Life Sciences, Henan University, Kaifeng 475004, China

**Keywords:** surrogate species, environmental risk assessment, transgenic maize, ELISA, non-target arthropods

## Abstract

While transgenic *Bacillus thuringiensis* (*Bt*) maize provides pest resistance and a reduced application of chemical pesticides, a comprehensive environmental risk assessment is mandatory before its field release. This research determined the concentrations of *Bt* protein in plant tissue and in arthropods under field conditions in Gongzhuling City, northeastern China, to provide guidance for the selection of indicator species for non-target risk assessment studies. *Bt* maize expressing Cry1Ab/2Aj and non-transformed near-isoline were grown under identical environmental and agricultural conditions. Cry1Ab/2Aj was detected in plant tissues and arthropods collected from *Bt* maize plots during pre-flowering, flowering, and post-flowering. The expression of Cry1Ab/2Aj varied across growth stages and maize tissues, as well as in the collected arthropods at the three growth stages. Therefore, representative species should be chosen to cover the whole growing season and to represent different habitats and ecological functions. *Dalbulus maidis* (Hemiptera: Cicadellidae), *Rhopalosiphum padi* (Hemiptera: Aphididae), *Heteronychus arator* (Coleoptera: Scarabaeidae), and *Somaticus angulatus* (Coleoptera: Tenebrionidae) are suitable non-target herbivores. *Propylea japonica* (Coleoptera: Coccinellidae), *Paederus fuscipes* (Coleoptera: Staphylinidae), *Chrysoperla nipponensis* (Neuroptera: Chrysopidae), and spiders are suggested predators. *Apis cerana* and *Apis mellifera ligustica* (both Hymenoptera: Apidae) represent pollinators and *Folsomia candida* (Collembola: Isotomidae) decomposers.

## 1. Introduction

Since the rapid advancements in biotechnology, genetically engineered (GE) crops have been created by inserting foreign genes to confer traits that cannot be obtained through conventional breeding methods [1]. Some GE crops produce crystal proteins (Cry proteins) derived from *Bacillus thuringiensis* Berliner (*Bt*), which can effectively control pests such as Lepidoptera or Coleoptera. *Bt* crops can benefit the environment and the economy by decreasing environmental pollution caused by chemical pesticides and by enhancing crop quality and yield [2]. However, the cultivation of *Bt* crops remains controversial due to concerns about the evolution of pest resistance and the high exposure of non-target organisms to *Bt* proteins throughout the season [3].

Assessing the impact of *Bt* crops on non-target organisms is an essential component of an environmental risk assessment (ERA) prior to commercial cultivation [4,5]. Non-target organisms encompass a range of arthropod species that provide critical ecological functions, such as biological control, pollination, and decomposition, as well as species of cultural or conservation value [6]. Because testing all arthropod species is impossible, suitable representative indicator species must be selected. The testing of a limited number of representative species under laboratory conditions can help to predict the safety of transgenic *Bt* crops for the majority of arthropod species in the field. While this approach has certain limitations, the concept of representative species (indicator species) has also been established in related fields, such as toxicity testing [7,8] and environmental monitoring [9,10,11,12]. 

The selection of representative arthropod species must fulfill three criteria:The chosen species should face a high level of exposure to exogenous insecticidal proteins expressed by transgenic plants.The chosen species should be among those most likely to be sensitive to exogenous insecticidal proteins expressed by transgenic plants.The chosen species should represent providers of ecological or cultural services relevant for the cropping system in the region of anticipated use.

To accurately assess the impact of *Bt* crops on non-target organisms, it is crucial to understand the spatiotemporal expression of the *Bt* proteins and the feeding behavior of arthropods in the field. This includes factors such as the insecticidal spectrum and mode of action of the insecticidal protein, as well as non-target organisms that are present in the field. Studies have shown the spatiotemporal differences in the expression of insecticidal proteins in transgenic *Bt* crops [13,14]. The insect resistance of *Bt* crops is directly linked to the content of the insecticidal protein in the plants. Different growth stages and plant parts have variable *Bt* protein expression levels and therefore potentially different insecticidal effects. Thus, it is crucial to know the concentration of *Bt* proteins in different growth stages and parts of plants, as well as the species composition and *Bt* protein content of arthropods in the field at different plant growth stages.

Based on the aforementioned information, knowledge of the arthropod community present in the field, including taxonomy, ecological roles, abundance, and intricate relationships, is crucial when selecting suitable representative organisms for laboratory-based non-target effects evaluations. Assessing the potential effects of *Bt* crops in off-crop habitats is also part of the environmental risk assessment but not the subject of the current study. The transgenic maize line Shuangkang 12-5 (SK12-5) [15,16] expresses a *cry1Ab*/*2Aj* fusion gene, providing highly effective resistance against the Lepidoptera pests *Ostrinia furnacalis* (Guenée) (Lepidoptera: Crambidae) and *Helicoverpa armigera* (Hübner) (Lepidoptera: Noctuidae) and an *EPSPS* gene [17]. The Cry1Ab/2Aj fusion protein has not been used so far in commercial GE maize lines, but Cry1Ab and Cry2A proteins have been commercialized in several countries, mainly North and South America, but not in China. SK12-5 maize was the first GE maize to have regulatory approval (safety certification) and may soon be grown commercially in China [18]. This study determined the exposure level of arthropods in the field and *Bt* protein concentrations in different plant parts, including the pollen, tassel, leaf, stem, and root of the transgenic maize line SK12-5, at three main growth stages (before flowering, during flowering, and after flowering). Organisms collected from the field were classified, and their *Bt* protein content was determined using enzyme-linked immunosorbent assays (ELISAs). The findings of this study will serve as a reference for selecting suitable surrogate species for laboratory-based studies to support the ERA of GE crops in China.

## 2. Material and Methods

### 2.1. Experimental Design

The study was conducted in 2018 at the experimental fields of the Jilin Academy of Agricultural Sciences, located in Gongzhuling City, northeastern China (43°19′ N, 124°29′ E). The experimental field had black soil typical of northeastern China, with an alkali solution nitrogen content of 77.5 ± 0.07 mg·kg^−1^, an organic matter content of 27.1 ± 0.07 mg·kg^−1^, a rapidly available potassium level of 154.1 ± 0.76 mg·kg^−1^, an available soil phosphorus level of 10.7 ± 0.07 mg·kg^−1^, and a pH value of 5.4 ± 0.02. The seeds of *Bt* maize (SK12-5) and its corresponding non-transformed near-isoline Lianchuang 303 (LC303, non-*Bt* maize) were used in this experiment. These maize lines were cultivated in adjacent plots measuring 50 m^2^ each, with three replicate plots allocated for each maize line. To reduce cross-contamination, adjacent plots were separated by a 2 m-wide bare dirt buffer strip. While pollen and insects were collected in all *Bt* plots, other samples were taken only from two of the three replicate plots. The field planting and management methods were described in more detail in a previous study [16]. The plots were cultivated using standardized agricultural management practices, and no insecticides were applied during the study.

### 2.2. Sample Collection

Plants, soil, and arthropods were sampled at various plant growth stages, including the ‘before flowering stage’ (BF), ‘during flowering stage’ (DF), and ‘after flowering stage’ (AF).

Leaf samples were collected as 3–4 cm fragments from two plants per plot in two plots of each maize line, randomly selected from the middle of the maize plants. Leaf sampling was conducted daily for seven days during the BF and AF stages and for two days during the DF stage due to time limitations associated with pollen collection during that period. The pollen collection from each plot followed the methodology outlined in [19]. Additionally, four entire plants were randomly chosen from two plots of each maize line at each stage (BF, DF, and AF). After rinsing the roots with water, samples of roots, stems, and tassels were collected. Overall, 102 samples were collected for each maize line (64 for leaves, 6 for pollen, 12 for roots, 12 for stems, and 8 for tassels). All samples were stored at −80 °C until they underwent ELISA processing.

To investigate the degradation of *Bt* protein in leaf material after entering the soil, a total of twenty-four 40-mesh nylon bags (25 cm × 15 cm, purchased from Xindongshan Agricultural Co., Ltd. in Meishan City, China) were deployed across two plots on 20 July 2018. Leaf samples from *Bt* maize weighing approximately 50 g were collected from plants within each plot and placed directly into individual mesh bags. These bags were randomly buried at a depth of 10 cm in various locations within the *Bt* maize field, with each bag’s position marked using an inserted tag. On 20 August, 20 September, 20 October, and 20 November, three mesh bags were randomly chosen and retrieved. All mesh bags were opened, and their contents were placed into ziplock bags and stored at −80 °C until subsequent processing using ELISAs.

For soil sampling, four points were randomly chosen from the “root zone” of maize plants on the ridge during the DF and AF stages in two plots of each maize line. Soil cores with a diameter of 10 cm and a depth of 15 cm were collected at each selected point. Approximately 1 kg of soil was collected for each sample and transferred to the laboratory in self-sealing bags. This resulted in a total of 8 samples for each maize line. All soil samples were stored at −80 °C until they were processed for ELISAs.

Non-target arthropods from the two maize lines were collected by sweep-netting, visual collection, and pitfall trapping using a sampling scheme following an “X” pattern that covered the whole plot. Members of the target order Lepidoptera were not collected. The collected arthropods were determined based on a taxonomic database. For details on sampling and taxonomic determination, see Yang et al. [16]. The collected arthropods were placed separately into centrifuge tubes and immediately frozen at −20 °C in a portable freezer (Alpicool ENX42, Foshan Alpicool Electrical Appliance Co., Ltd., Foshan city, China). All captured arthropods were taken to the laboratory, placed in Petri dishes over dry ice, and examined using a Zeiss stereomicroscope (Carl Zeiss Digital Innovation GmbH., Dresden, Germany) for taxonomic identification. After identification to the species or family level, the samples were stored at −80 °C. 

### 2.3. ELISA Measurements

To ensure the accuracy of the measurements, phosphate-buffered saline Tween (PBST) buffer was used to remove surface pollutants from the leaf samples, stem samples, root samples, and arthropod samples. The Cry1Ab/2Aj fusion protein content in the maize tissue, soil, and arthropods was determined using a Cry1Ab/Ac ELISA kit from EnviroLogix (Portland, ME, USA). After freeze-drying, the samples were weighed. Samples of maize leaves, stems, tassels, roots, and soil were mixed with 1000 μL of PBST, and samples of maize pollen, leaves in mesh bags, and arthropods were mixed with 500 μL of PBST in 2.0 mL or 5.0 mL centrifuge tubes. Maize pollen, tassels, and arthropods were fully mazerated by hand using an electric grinding rod (Jingxin MY-10, purchased from Shanghai Jingxin Industrial Development Co., Ltd., Shanghai, China) on ice. Maize leaves, stems, roots, and maize leaves in mesh bags were ground using a high-throughput cryogenic tissue homogenizer (QM 100, purchased from Wuzhou Dingchuang Technology Co., Ltd., Beijing, China). After centrifugation and the appropriate dilution of the supernatants (1000 times dilution for maize leaves and pollen, 100 times for maize stems, tassels, roots, and soils, no dilution for maize leaves in mesh bags and arthropods, and for samples from non-*Bt* maize), six concentrations of purified Cry1Ab protein (EnviroLogix, Portland, ME, USA) with a purity of >95% were loaded twice on each plate for constructing a standard curve. The ELISA was performed following the manufacturer’s instructions. The optical density (OD) values were measured using a microplate spectrophotometer (Power Wave XS2, BioTek, Santa Clara, CA, USA).

### 2.4. Statistical Analysis

A standard curve was established using a single rectangular hyperbola model, and the concentrations of Cry1Ab/2Aj in the maize tissues, soil, or arthropods were calculated by comparing the OD values to the standard curve, considering the sample dry weight, the amount of added extraction buffer, and the dilution. The reported Cry1Ab/2Aj concentrations should be considered “Cry1Ab-equivalents” because Cry1Ab served as standards and not the Cry1Ab/2Aj fusion protein that is produced by the plants. The limit of detection (LOD) for the Cry1Ab/2Aj protein in the ELISA was calculated by determining the standard deviation of all blank values from 5 ELISA plates. Three times this standard deviation was then considered the LOD concentration, which was 0.015 ng/mL for this study. In the following, the lowest and highest values of the 95% confidence intervals (95CIs) were used for the means of each maize material or arthropod taxa, and differences were considered significant if the 95CIs did not overlap. In addition, medians were calculated and used for figures. Arthropod species with less than 3 samples per growth stage were omitted for further calculations as those samples would not be reliable.

## 3. Results

### 3.1. Cry Protein in Maize Tissues

The ELISA assays conducted on maize tissues at three different growth stages indicated that the concentrations of Cry1Ab/2Aj in the roots were highest in the AF stage, and the contents were significantly higher (non-overlapping 95CIs) in the DF stage than in the BF stage (non-overlapping 95CIs, Figure 1). The concentrations of Cry protein in the tassels were significantly higher in the DF stage than in the AF stage (Figure 1). There were no differences in the Cry protein contents in the maize leaves or stems for the three growth stages (Figure 1). 

For the BF stage, the concentrations of Cry1Ab/2Aj in leaves were highest, and the concentrations were higher in the stems than in the roots (Figure 1). In the DF stage, the concentrations in the leaves were significantly higher than in the other maize tissues (pollen, tassels, stems, and roots), and the concentrations in the pollen were significantly higher than in the tassels, stems, or roots, while there were no significant differences for the other maize tissues (Figure 1). For the AF stage, the Cry protein concentrations in the leaves were significantly higher than in the other maize tissues (tassels, stems, and roots). The concentrations in the roots were higher than in the tassels or stems, and the concentrations in the stems were significantly higher than in the tassels (Figure 1). No Cry protein was detected in the non-transformed near-isoline maize LC303 tissue.

### 3.2. Degradation of Bt Maize Leaves in Soil

Compared to the concentrations of Cry1Ab/2Aj in the maize leaves collected from the plants in July, the concentrations measured in the samples recollected in the following months from the mesh bags that were buried in the soil were significantly lower (non-overlapping 95CIs). The concentrations of *Bt* protein detected in the samples from November were below the limit of detection (<0.020 µg/g). No significant differences in the Cry protein concentrations of the maize leaves in the bags collected from August to November were observed (Figure 2). 

### 3.3. Cry Protein in Soil

In the DF stage, two of the four collected soil samples showed very low Cry1Ab/2Aj concentrations (≤0.0020 µg/g), while the other two samples were below the LOD (0.0015 µg/g). In the AF stage, two of the four samples also had low Cry1Ab/2Aj concentrations (≤0.0025 µg/g), while the other two samples were below the LOD (0.0018 µg/g).

### 3.4. Exposure of Arthropods to Cry1Ab/2Aj Produced by Bt Maize

In the BF stage, a total of 177 arthropods from over 31 taxa, including herbivores, decomposers, parasitoids, pollinators, predators, and others, were collected, and their Cry protein contents were measured (Figure 3). The highest concentrations of Cry1Ab/2Aj were detected in the groups of herbivores, followed by predators and others (Diptera). Among the herbivores, the highest contents were found in Coleoptera and Hemiptera. *Trigonotylus ruficornis* (Geoffroy) (Hemiptera: Miridae) contained the highest concentrations among all the collected arthropods (0.37 μg/g DW), followed by *Monolepta typographica* (Weise) (Coleoptera: Chrysomelidae) (0.13 μg/g) and *Adrisa magna* (Uhler) (Hemiptera: Cydnidae) (0.011 μg/g). Among predators, the Araneae and Coleoptera showed the highest Cry protein contents. *Clubiona interjecta* (Koch) (Araneae: Clubionidae) (0.021 μg/g), *Clubiona japonicola* Bösenberg and Strand (Araneae: Clubionidae) (0.035 μg/g), *Gnathonarium dentatum* (Wider) (Araneae: Linyphiidae) (0.032 μg/g), *Hylyphantes graminicola* (Sundevall) (Araneae: Linyphiidae) (0.020 μg/g), *Xysticus ephippiatus* Simon (Araneae: Thomisidae) (0.013 μg/g), and *Paederus fuscipes* Curtis (Coleoptera: Staphylinidae) (0.059 μg/g) contained more Cry protein than the other collected predators. In the group “others”, *Musca domestica* Linnaeus (Diptera: Muscidae) (0.024 μg/g), *Ophyra nigra* (Wiedemann) (Diptera: Muscidae) (0.014 μg/g), and *Sarcophaga melanura* Meigen (Diptera: Sarcophagidae) (0.036 μg/g) showed comparatively high Cry protein concentrations.

In the DF stage, a total of 277 arthropods, distributed over 36 taxa, were collected (Figure 3). For herbivores, the highest detected contents were in Hemiptera, Diptera, and Coleoptera. *Trigonotylus ruficornis* (0.30 μg/g) and *Drosophila macquarti* (Wheeler) (Diptera: Drosophilidae) (0.20 μg/g) contained higher concentrations than the other collected herbivores. Significant amounts of Cry1Ab/2Aj (0.01–0.1 µg/g) were detected in *Aeoloderma agnata* (Candeze), *Melanotus caudex* (Lewis) (both Coleoptera: Elateridae), *M. typographica*, *Oscinella frit* (Linnaeus) (Diptera: Chloropidae), and *Sitobion avenae* (Fabricius) (Hemiptera: Aphididae). For the decomposers, Isotomidae (Entomobryomorpha) contained around 0.031 μg/g. For the group of parasitoids, Ichneumonidae (0.086 μg/g) contained more Cry protein than the other collected taxa. During the flowering stage, hymenopteran pollinators like *Apis cerana* Fabricius (Hymenoptera: Apidae) (0.018 μg/g) and Vespidae sp. (Hymenoptera) (0.010 μg/g) also contained significant amounts of Cry protein. For predators, Araneae, Coleoptera, Hemiptera, and Hymenoptera contained the highest *Bt* protein concentrations. *Hylyphantes graminicola* (0.071 μg/g), *Lycosa coelestis* Koch (Araneae: Lycosidae) (0.036 μg/g), *Scymnus hoffmanni* Weise (Coleoptera: Coccinellidae) (0.046 μg/g), *P. fuscipes* (0.036 μg/g), *Orius strigicollis* (Poppius) (Hemiptera: Anthocoridae) (0.067 μg/g), *Cyrtorhinus lividipennis* Reuter (Hemiptera: Miridae) (0.015 μg/g), *Nabis stenoferus* Hsiao (Hemiptera: Nabidae) (0.027 μg/g), and Vespidae sp. (0.010 μg/g) contained more Cry protein than the other collected predators. In the group of “others”, *M. domestica* contained significant amounts of Cry protein (0.085 μg/g).

In the AF stage, a total of 127 arthropods, distributed over 28 taxa, were collected (Figure 3). The highest detected *Bt* protein concentrations for herbivores were in *Rhopalosiphum maidis* (Fitch) (Hemiptera: Aphididae) and *D. macquarti* (0.01–0.1 μg/g). For the predators, *H. graminicola* (0.021 μg/g), Tabanidae (Diptera) (0.044 μg/g), and Hemerobiidae (Neuroptera) (0.012 μg/g) contained more Cry protein than the other collected predators. For the group of “others”, Ceratopogonidae (0.016 μg/g) and *M. domestica* (0.059 μg/g) showed comparatively high Cry protein concentrations.

Species for which less than three samples were available per growth stage and maize line are listed in the Appendix A (data table).

## 4. Discussion

For the assessment of the exposure of non-target species to the Cry proteins in the ERA, the spatiotemporal expression of *Bt* protein in the plant and the concentrations of *Bt* protein in the arthropod communities inhabiting *Bt* crops are important pieces of information. In this study, the production of Cry1Ab/2Aj protein in the *Bt* maize tissues varied in different growth stages and different plant parts. The maize leaves generally contained more *Bt* protein than the other tissues, i.e., the stems, roots, tassels, and pollen. There was no clear difference in Cry protein concentrations between the growth stages for the leaves and stems, while the Cry protein content in the tassels decreased, and in the roots increased after flowering. Obviously, during the flowering period, the pollen, as a route of exposure, was highly abundant, as maize is wind-pollinated and many arthropods, including pollinators, are known to consume maize pollen [20]. The Cry1Ab/2Aj-expressing maize had substantial amounts of the insecticidal protein in pollen, with concentrations of approximately half of those in leaves.

The highest detected contents in the arthropods collected in the field were approximately two orders of magnitude lower than in the leaves, and the concentrations changed over the season. For all three maize stages, the herbivores contained the highest amounts of Cry proteins, in particular certain Hemiptera and Coleoptera species, which are known to feed directly on maize leaves or pollen, like *T. ruficornis* and *M. typographica*. It is known that Cry proteins are excreted and digested quickly, causing a reduction in Cry protein concentrations from lower to higher trophic levels. This coincides with previous research, including Cry1Ab maize [21,22], Cry3Bb1 maize [23], Cry2A rice [24], Cry1Ac cotton [25], Cry1Ac and Cry2Ab cotton [14], and Cry1Ac soybean [26]. Among herbivores, particularly within the Hemiptera, the amount of *Bt* protein differs according to different feeding modes. Although all Hemiptera have piercing-sucking mouthparts, some species, including stink and plant bugs (Heteroptera), extract sap from green plant tissue, while others, such as aphids and some leafhoppers, feed on phloem sap. Consequently, bugs often contain high levels of Cry protein, while phloem-feeding aphids and leafhoppers contain no or only trace amounts [27]. Nevertheless, the individual values of some phloem-feeding species (*S. avenae*, *R. maidis*) were well above the detection limit, which might indicate artefacts or the contamination of samples with plant tissue, which is particularly likely during the flowering period. Regarding the effects of Cry1Ab/2Aj on herbivores, data are available only for silk worms (*Bombyx mori* Linnaeus, Lepidoptera: Bombycidae), which are economically important insects belonging to the target order of the *Bt* protein (Lepidoptera). When the concentration of transgenic maize pollen reached 10,000 grains/cm^2^ of mulberry leaves (approximately 38 μg/g), a lethal effect was observed [28,29,30]. Such doses are highly unrealistic in the field and approximately 100–4000 times higher than our detection in maize fields. Nevertheless, non-target butterflies and moths, which may exist in semi-natural vegetation surrounding maize fields, should be included in the risk assessment of Cry1Ab/2Aj-producing maize, in particular because this maize has relatively high *Bt* protein concentrations in pollen, and Lepidoptera belong to the target order of this *Bt* protein. However, practical and standardized selection procedures have hardly been developed and applied for Cry1Ab/2Aj maize and non-target Lepidoptera so far. We can refer to the assessments for Cry1Ab and non-target Lepidoptera, where some studies took non-target butterflies, i.e., *Danaus plexippus* (Linnaeus) [31], *Aglais urticae* (Linnaeus) [32], both Nymphalidae; *Papilio polyxenes* Fabricius (Papilionidae) [33] and *Antheraea pernyi* (Guerin-Meneville) Saturniidae [34] as representative species and showed no negative effects. However, the commercial Cry1Ab line (MON810) has very low concentrations in pollen compared with the SK12-5 maize line. 

The risk assessment follows a tiered approach that starts with laboratory studies under worst-case exposure conditions. While it is not possible to test all species that are potentially present in the environment, surrogate species are selected that represent different habitats or different ecological functions [35]. Based on their abundance in Chinese maize fields [16] and their exposure to Cry protein, *T. ruficornis* and *M. typographica* would be suitable candidates as Hemiptera or Coleoptera non-target herbivore species. While there are some available rearing protocols for omnivorous mirids, rearing methods or testing protocols for phytophagous mirids, like *T. ruficornis*, are lacking. For *M. typographica*, it is difficult to separate their eggs from the soil, and the species is not suitable for large-scale breeding in a laboratory. Due to these reasons, several studies have selected *Dalbulus maidis* (DeLong) (Hemiptera: Cicadellidae) [36], *Rhopalosiphum padi* (Linnaeus) (Hemiptera: Aphididae) [37], *Heteronychus arator* (Fabricius) (Coleoptera: Scarabaeidae) [38], and *Somaticus angulatus* (Fåhraeus) (Coleoptera: Tenebrionidae) [38] as alternative non-target herbivores to determine the effects of GE crops in laboratory studies.

The risk assessment of transgenic *Bt* crops should focus on taxonomic and functional groups that are both common and highly exposed to Cry proteins produced from GE crops [39]. In our previous study, the proportions of non-target arthropods collected in *Bt* maize fields revealed that Hemiptera accounted for more than 50% of the collected arthropods during the BF and DF stages, while Coleoptera accounted for more than 50% during the AF stage [16]. In our current study, we found that certain taxa within the herbivore and predator groups exhibited comparatively high Cry protein contents. It is important to note that predators may be exposed to Cry protein through their consumption of prey as well as pollen or leaf tissues as supplementary food sources. Our results indicate that many predator species, including Dermaptera, Diptera, Coleoptera, Hemiptera, and Neuroptera, had comparatively high Cry protein levels, although these concentrations were approximately three orders of magnitude lower than in the maize tissue. Predatory arthropods are commonly recommended as surrogate test species for supporting the ERA of insecticidal GE maize. Based on their abundance in Chinese maize fields [16], exposure to Cry protein, and suitability for testing under laboratory conditions, the following predators can be suggested as surrogate species: *Propylea japonica* (Thunberg) (Coleoptera: Coccinellidae), *P. fuscipes*, *C. nipponensis*, and spiders. Previous ERA studies have already utilized these species as surrogates, including *P. japonica* [40,41,42], *P. fuscipes* [43], *C. nipponensis* [44], and the spider *Ummeliata insecticeps* (Bösenberg and Strand) (Aranea: Linyphiidae) [45]. There were some studies on the effect of maize materials containing Cry1Ab/2Aj protein on non-target predators. No negative effects were observed on *P*. *japonica* (feeding dose up to 3.17 μg Cry1Ab/2Aj/g pollen) [42], *Chrysoperla sinica* (Okamoto) (Neuroptera: Chrysopidae) (up to 2.06 μg/g Cry1Ab/2Aj detected in the body) [44], and *Harmonia axyridis* (Pallas) (Coleoptera: Coccinellidae) (feeding dose up to 3.16 μg/g *Spodoptera exigua* detected in the body) [28,46]. 

In this experiment with Cry1Ab/Cry2Aj-producing maize, pollen contained comparatively high amounts of Cry protein during the flowering stage (ca. half of the concentration in leaves), and the collected pollinators during this stage also showed relatively high Cry protein contents, around three orders of magnitude lower than in pollen and leaves. Therefore, pollinators should be included in the non-target testing for Cry1Ab/2Aj-producing maize. Based on our research, *A. cerana* can be suggested as a pollinator surrogate due to the fact that they are abundant in Chinese maize fields [16], exposed to Cry protein, and available for testing under laboratory conditions. Published studies not only applied *A. cerana* [47,48] but also *Apis mellifera ligustica* Spinola (Hymenoptera: Apidae) [49,50] as pollinator surrogates in ERA. There were no negative effects of Cry1Ab/2Aj on *A. mellifera* adults (feeding dose up to 728.78 ng Cry1Ab/2Aj [51] and up to 0.728 μg/g Cry1Ab/2Aj detected in the body) [29]. Pollinators can directly come into contact with the pollen produced by transgenic *Bt* crops in the field and bring it back to their colonies, which may have an impact on populations and biodiversity [49,52]. Differences in pollen quality, flower odor, appearance, and other plant characteristics can also affect the collection behavior of pollinators. Therefore, when evaluating the safety of transgenic plants with high *Bt* protein contents in pollen for pollinators, monitoring populations and species diversity under natural conditions may be carried out in addition to direct feeding assays with insecticidal proteins under laboratory conditions. In addition to the potential effects of GE crops on non-target species, outcrossing is another area of concern when assessing the risks of GE crops. In the case of maize, pollinators and pollen feeders may carry GE maize pollen to non-GE maize fields. However, compared to insect-pollinated crops, the amount of pollen that is deposited on the silk can be considered low because maize is wind-pollinated, with anthers and silk being spatially separated. Consequently, bees and other pollinating species are not expected to contribute to a great extent to maize pollination and thus to the presence of *Bt*/non-*Bt* hybrid kernels that may further expose non-targets in non-*Bt* fields. Cross-fertilization events between *Bt* and non-*Bt* maize are more likely driven by maize pollen drifting by wind, although this route of exposure is mainly limited by the relatively short distance of pollen deposition within and around the maize field and can be managed by implementing isolation distances between *Bt* and non-*Bt* fields [53]. Also, the outcrossing of GE maize to wild species is not of concern in China because maize has no compatible relatives in China [54].

Although the Cry protein contents in the soil were close to or below the LOD of the ELISA assay, the degradation study of maize leaves revealed that Cry protein contents could still be detected after three months, albeit at very low concentrations. In farmland ecosystems, *Bt* proteins can be released into the soil through various pathways, such as leachates from pollen and straw. In our study, we observed the presence of Cry protein (in low doses) in decomposer species. Among the decomposer species, the Isotomidae family stands out as an indicator group due to its high abundance in Chinese maize fields [16] and exposure to Cry protein within these fields. Within the Isotomidae, *Folsomia candida* Willem (Collembola: Isotomidae) emerges as a suitable species, given its wide distribution across different ecosystems. *F. candida* has been extensively used as a model organism for evaluating the impact of pesticides on soil organisms over a significant period of time. Therefore, when evaluating the effects of GE crop cultivation on beneficial soil organisms, this species is usually included in safety testing portfolios. Previous studies have reported no negative effect of transgenic cry1Ab/2Aj maize on *F. candida* (up to 464.05 ng/g Cry1Ab/2Aj protein detected in the body) [55]. Researchers have used *F. candida* [56], as well as *Xenylla grisea* Axelson (Collembola: Hypogastruridae) [57], *Entomobrya griseoolivata* (Packard) (Collembola: Entomobryidae), and *Bourletiella christianseni* Snider (Collembola: Bourletiellidae) [58], in GE plant ERA studies. 

## 5. Conclusions

To evaluate the safety of insect-resistant transgenic crops on non-target organisms, it is not feasible to test every arthropod species in the field due to their vast diversity. Therefore, representative species for testing must be selected. To better predict the potential negative impacts of planting insect-resistant transgenic crops on non-target organisms, species that are potentially sensitive to the protein and have a high exposure level should be selected for evaluation. Moreover, because the expression of *Bt* protein varies during different stages of plant growth, representative species should be chosen to cover the whole growing season and to represent different habitats and ecological functions. Based on the selection criteria for representing arthropod species, including high exposure and potential sensitivity to the insecticidal protein and abundance in Chinese maize fields [16], as well as practical considerations, such as the availability of the species and testing protocols, and the presence of economic or ornamental values, *D. maidis*, *R. padi*, *H. arator*, and *S. angulatus* are suitable non-target herbivores. *Propylea japonica*, *P. fuscipes*, *C. nipponensis*, and spiders are suggested predators. *Apis cerana* and *A. mellifera ligustica* represent pollinators, and *F. candida* represents decomposers. In addition, surrogate species should also represent significant species of conservation importance in a region, like the silk worm *B. mori*. Additionally, significant differences in community composition exist across different geographical regions. Therefore, establishing a biological information database for species populations in various regions can also assist in selecting representative arthropod species in the field. In conclusion, this study provides evidence on the exposure of various arthropod species, among which representatives should be selected for future safety evaluations of GE maize.

## Figures and Tables

**Figure 1 plants-12-04057-f001:**
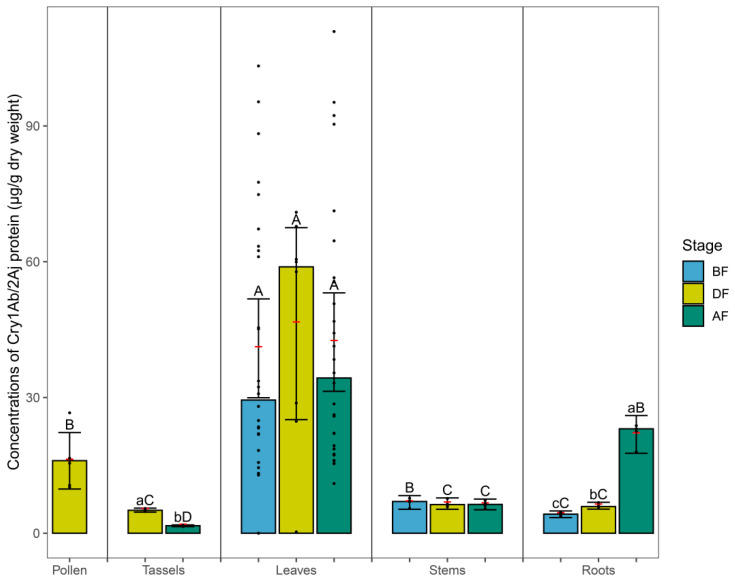
Cry1Ab/2Aj protein concentrations (μg/g dry weight) in the same stage of different maize tissues: leaves, stems, and roots before flowering (BF); pollen, tassels, leaves, stems, and roots during flowering (DF); tassels, leaves, stems, and roots after flowering (AF). Bars represent medians, and red lines with error bars represent means and 95CIs for each stage (*n* = 6 for pollen; *n* = 28, 8, 28 for leaves BF, DF, and AF, respectively; *n* = 4 for tassels, stems, and roots in each stage). Different small letters indicate significant differences between maize stages, and capital letters indicate significant differences between tissue types based on non-overlapping CIs. Black dots represent the individual values.

**Figure 2 plants-12-04057-f002:**
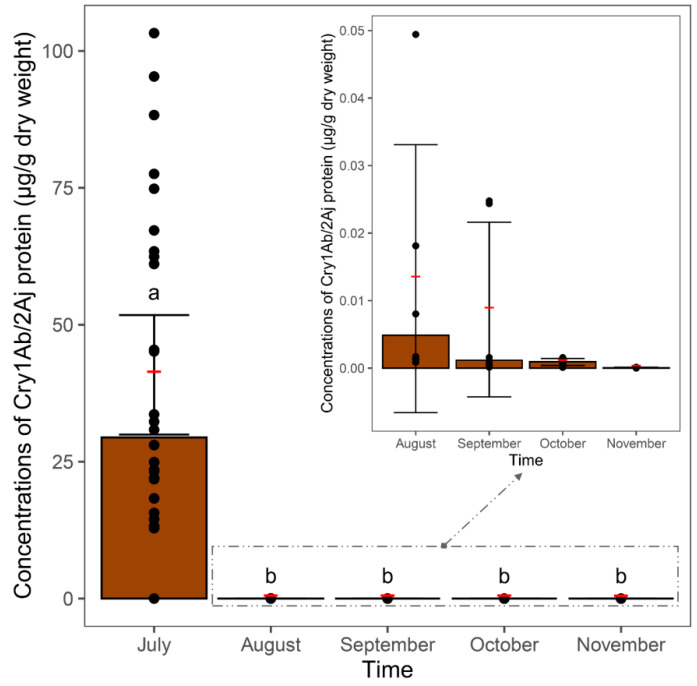
Concentrations of Cry1Ab/2Aj protein (μg/g dry weight) in fresh maize leaves (July sample) and in maize leaves buried in soil for 4 months (August–November samples). Bars represent medians, and red lines with error bars represent means and 95CIs for each month (*n* = 6 for each month). Different letters indicate significant differences. Black dots represent individual values.

**Figure 3 plants-12-04057-f003:**
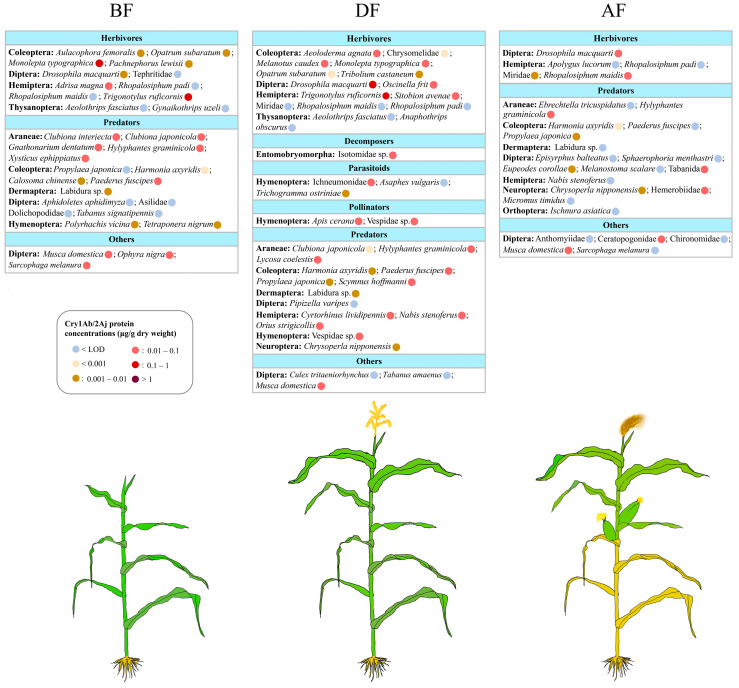
Cry1Ab/2Aj protein concentrations (μg/g dry weight) in each collected arthropod species for 3 maize stages (BF: before flowering; DF: during flowering; AF: after flowering). Species are grouped into orders. The colored circles following arthropod species or families indicate the detected median concentrations.

## Data Availability

All ELISA data presented in this manuscript are available in the online Appendix A.

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
