# Peer review of "Expression of Cry1Ab/2Aj Protein in Genetically Engineered Maize Plants and Its Transfer in the Arthropod Food Web"

_plants, 2023, doi:10.3390/plants12234057_

Round 1
Reviewer 1 Report
Comments and Suggestions for Authors
Manuscript ID: plants-2694539 titled “Expression of Cry1Ab/2Aj Protein in Genetically Engineered Maize Plants and its Transfer in The Arthropod Food Web” by Yi Chen et al. submitted to section: Plant Protection and Biotic Interactions, is a well written manuscript that flows well from start to end with a good discussion and literature citation. Authors provided important results and protocols for risk assessment to select representative indicator species that can predict the safety of transgenic Bt crops for the most arthropod species in Maize crop in China. I added some editorial notes to the attached PDF. In my opinion, this article should be considered for publication pending minor revision. My concern is:
· Missing in material and methods how all taxa were determined, for reliable identification and where vouchers are deposited?

Author Response
Thank you very much for your positive comments on our manuscript and the details in the PDF document. Please find the detailed responses below and the corresponding corrections highlighted in the re-submitted file. We agree with most of the comments in the PDF documents. Therefore, we have made the corrections according to your suggestions. Changes are marked in red.
Line 123 in PDF version (correspondent line 134 in revised manusript) “Overall, 102 samples were collected for each maize line (64 for leaves, 6 for pollen, 12 for roots and stems, and 8 for tassels).” Your suggestion is the total number was 64+6+12+8, but we had 12 samples for roots and 12 samples for stems, which was 64+6+12+12+8. We made this clear in the revised text.
Line 390, 409, 416, and 417 in the PDF version (corresponding to lines 428, 448, 456, and 458, respectively, in the revised manuscript), you provided comments regarding the Latin species authorship. We have revised it in accordance with your suggestions.
For your concern regarding taxonomic identification, we have written the details in our related published paper “Yan Yang, Yi Chen, Jiabao Xue, Yuanyuan Wang, Xinyuan Song, Yunhe Li. Impact of Transgenic Cry1Ab/2Aj Maize on Abundance of Non-Target Arthropods in the Field. Plants, 2022, 11(19): 2520. https://doi.org/10.3390/plants11192520”. Thanks for pointing this out, we have added some simple descriptions in line 154.
Reviewer 2 Report
Comments and Suggestions for Authors
The article submitted for review concerns Cry1Ab/2Aj protein expression in genetically modified maize and its transfer into the arthropod food web. The authors determined the expression levels of Cry protein in different parts of crop plants at different periods of plant growth. The topic is interesting because the cultivation of GM maize, however it reduces the amount of insecticides used, raises environmental risks. This risk, should be determined in relation to locally occurring arthropod communities. Generally, the article was well written and can be accepted for publication. However, my objections are raised by the rather limited study material, the extremely different size of the study groups (from 6 to 64) and the way the statistical analyses were conducted.
Author Response
Thanks for your time and comments. The sample numbers for our different groups are different, from 6 to 64. In detail, n = 6 for pollen; n = 28, 8, 28 for leaves at BF, DF, and AF, respectively; n = 4 for tassels, stems, and roots in each stage. Our purpose for collecting those maize tissues was to test for the Cry protein contents based on ELISA assays, for which we believe that a minimum of four samples was enough for data accuracy. We provide data of different tissues and different growth stages and the number of samples that we could process was limited. For leaves, which showed the highest variability, we analyzed also a higher number of samples than for the other tissues, which were more uniform. The lowest and the highest values of the 95% confidence intervals (95CIs) were used for the means of each group, and differences were considered significant if the 95CIs did not overlap. Because of the unbalanced number of samples we decided to work with this basic statistical method rather than applying more sophisticated statistical tools. The differences that we wanted to detect in the ELISA data are more in the order of magnitude level and the method of comparing 95CIs is sufficient for that.
Reviewer 3 Report
Comments and Suggestions for Authors
The article “Expression of Cry1Ab/2Aj Protein in Genetically Engineered Maize Plants and its Transfer in the Arthropod Food Web” (plants-2694539) studies the expression of Cry1Ab/2Aj protein in GMO maize and its content on different arthropods. The research fits the journal, but some things are unclear, so clarifications and further explanations are needed.
Introduction
The authors should write about the use of Cry1Ab/2Aj for insect resistance in maize and/or other crops.
The authors should also provide further information on why the tested non-target arthropod species were selected and how they fulfill the criteria indicated in lines 64-70. This information should either be added in the Introduction or in the Discussion.
Material and Methods, section 2.2
Line 118: it is not clear what you mean by “samples were collected as 12 for stems cm fragments”.
Lines 118-119: by “two plants in two plots”, I guess you mean one plant in each of two plots, is this correct?
Lines 118-122: could the concentrations of Cry1Ab/2Aj protein on leaves during flowering be affected by the concentration of Cry1Ab/2Aj protein in pollen due to pollen deposition on leaves?
Lines 144-145: More information should be provided on the collection of arthropods. For example, where were the pitfall traps placed in the plot? Was collection of arthropods only conducted in the plots with Bt maize (SK12-5)?
Results
Line 200, legend of Figure 1: do the error bars refer to standard error or standard deviation?
Line 268: does the Vespidae (Hymenoptera) refer to only one unidentified species of Vespidae?
Lines 285-286: this sentence should be in the Materials and Methods, section 2.2.
Lines 286-310: this part seems unnecessary in the text and it could be added as supplementary material.
Discussion
As mentioned above, the authors should discuss why the tested non-target arthropod species fulfill the criteria indicated in lines 64-70 of the text.
Given the content of Cry1Ab/2Aj protein in pollinators, the authors should discuss how these could affect contamination in neighboring non-Bt crops. The authors should also discuss the potential effect of the Bt toxins on non-target Lepidoptera that are not economic pests in maize.
Supplementary material
On the datasheet of maize material, if you use the abbreviation Con, indicate that it refers to concentration and also specify the concentration units. Authors should also indicate that the abbreviation LOD refers to limit of detection.
Round 2
Reviewer 3 Report
Comments and Suggestions for Authors
The manuscript has been improved. However, my suggestion regarding Bt contamination on neighboring non-Bt crops should be further explained. What the authors write in lines 405-408 of the Discussion section requires further explanation and references.
